# Characteristics of Metabolites in the Development of Atherosclerosis in Tibetan Minipigs Determined Using Untargeted Metabolomics

**DOI:** 10.3390/nu15204425

**Published:** 2023-10-18

**Authors:** Liye Shen, Jinlong Wang, Yongming Pan, Junjie Huang, Keyan Zhu, Haiye Tu, Minli Chen

**Affiliations:** 1Academy of Chinese Medicine & Institute of Comparative Medicine, Zhejiang Chinese Medical University, Hangzhou 310024, China; shenliye@ucas.ac.cn (L.S.); pym@zcmu.edu.cn (Y.P.); h306999492@126.com (J.H.); zjzygyz@163.com (K.Z.); tuhaiye1999@163.com (H.T.); 2Key Laboratory of Systems Health Science of Zhejiang Province, Hangzhou Institute for Advanced Study, University of Chinese Academy of Sciences, Chinese Academy of Sciences, Hangzhou 310024, China; jinlongwang@ucas.ac.cn

**Keywords:** atherosclerosis, Tibetan minipig, untargeted metabolomics

## Abstract

Background: Atherosclerosis (AS) is a chronic progressive disease caused by lipometabolic disorder. However, the pathological characteristics and mechanism of AS have not been fully clarified. Through high-fat and high-cholesterol diet induction, Tibetan minipigs can be used as the AS model animals, as they have a very similar AS pathogenesis to humans. Methods: In this study, we built an AS model of Tibetan minipigs and identified the differential abundance metabolites in the development of AS based on untargeted metabolomics. Results: We found that sphingolipid metabolism and glucose oxidation were obviously higher in the AS group and phenylalanine metabolism was reduced in the AS group. Moreover, in the development of AS, gluconolactone was enriched in the late stage of AS whereas biopterin was enriched in the early stage of AS. Conclusions: Our research provides novel clues to investigate the metabolic mechanism of AS from the perspective of metabolomics.

## 1. Introduction

Cardiovascular disease (CVD) is a leading cause of mortality worldwide and is a major contributor to reducing patients’ quality of life [1]. The number of deaths due to CVD has increased steadily, from 12.1 million in 1990 to 18.6 million in 2019, accounting for 1/3 of the total number of deaths globally [2]. China is one of the countries with a high burden of CVD. The number of deaths from CVD accounts for 40% of the total deaths, among which, the number of deaths from atherosclerosis cardiovascular diseases (ASCVD) accounts for 61% of the total deaths from CVD. In the past 20 to 30 years, the incidence of ASCVD has been rising, causing a huge economic burden on the country’s medical and health systems [3].

Atherosclerosis (AS) is a chronic progressive disease caused by lipometabolic disorder and chronic low-grade inflammation. It is the result of complex environmental and genetic factors [4,5]. Several studies have shown that genetic variation only accounts for a small part of the risk of developing AS, and environmental factors play a prominent role in the pathogenesis of AS [6]. Diet is the most important environmental factor, and metabolic abnormalities caused by changes in diet structure are one of the main reasons for the development of AS [7]. Long-term consumption of a high-fat, high-cholesterol, and high-sugar diet increases the risk of AS [8].

The occurrence and development of AS are involved in the accumulation of vascular lipids, inflammation, oxidative stress, etc. In the early stage of AS lesions, endothelial cells are activated by various injury factors such as intravascular free radicals and abnormal lipid accumulation, thus leading to functional disorders, mainly in the form of lipid metabolism disorders. Low-density lipoprotein (LDL) is oxidized to ox-LDL by series of oxidation reactions. Ox-LDL binds to the endothelial cell surface of lectin-like oxidized low-density lipoprotein receptor 1 (LOX-1) to enter endothelial cells and activate them to produce an inflammatory cascade, such as through activation of the Toll-like receptor pathway and arachidonic acid pathway and the interplay between lipid metabolism disorder and the inflammatory response promotes atherogenic development [9,10].

The emergence of metabolomics has provided a new direction for the study of the pathological mechanism of AS. The differences in the levels of identified metabolites (such as amino acids, lipids, bile acids, etc.) are closely related to the occurrence and development of AS [11]. These small molecule metabolites have the potential to become biomarkers of cardiovascular diseases and provide a new direction for the prevention and treatment of atherosclerosis.

Tibetan minipigs are characteristic experimental animals in China. The organ systems of Tibetan minipig and human beings are not only similar in morphology, but also basically the same in terms of physiological function, especially the cardiovascular system, lipid metabolism structure, and AS lesion site. Therefore, they are currently recognized as suitable model animals for the study of AS [12]. In our previous study, we found that Tibetan minipigs can easily develop AS under the induction of a high-fat diet, and a variety of risk factors for the formation of AS could be observed [13,14]. This AS model of central obesity is accompanied by insulin resistance and hypertension characterized by chronic inflammation and lipid metabolism disorder and is very similar to the etiological and pathological characteristics of human AS formation [15]. Therefore, in this study, the AS model of Tibetan minipigs was adopted to study the characteristics of the serum metabolites during the occurrence and development of AS, screen the relevant differential metabolites and their related pathways, and explain the relevant metabolic mechanism during the occurrence and development of AS.

## 2. Materials and Methods

### 2.1. Laboratory Animal

Twelve conventional male Tibetan minipigs, aged 4–5 months, and weighing 8–12 kg, were purchased from Dongguan Songshan Lake Pearl Laboratory Animal Technology Co., Ltd. (SCXK (Yue) 2017-0030) (Dongguan, China) under Certificate No. 44410500000286. All Tibetan minipigs were raised in the ordinary environment of the miniature pig laboratory of the Animal Experimental Research Center of Zhejiang Chinese Medical University (SYXK (Zhe) 2018-0012)(Hangzhou, China), at an ambient temperature of 22 ± 1 °C, with ambient relative humidity of 40–60%, and alternating light and dark for 12 h/12 h. The pigs were fed freely and were fed two meals a day. After 1 month of adaptive feeding in the laboratory, 6 animals were randomly assigned to each group according to body weight and blood biochemical indexes, and there was no significant difference between the two groups. All feeding and animal experiment procedures of this study were approved by the Laboratory Animal Management and Use Committee of the Animal Experimental Research Center of Zhejiang Chinese Medical University and the research team strictly ensured the welfare of the laboratory animals (IACUC approval number: 20191021-10).

### 2.2. Animal Serum and Aortic Vessel

The AS model group (model group) was fed with a high-fat and high-cholesterol (HFC) diet (HFC feed formula: 15% shortening, 10% yolk powder, 1.5% cholesterol, 0.5% choline, 73.0% basal diet) and the normal control group (NC group) was fed with a 100% basal diet. After 4, 8, 16, and 28 weeks, and after fasting for 16 h, the AS model group was compared with the normal control group at 28 weeks; 5 mL of blood was collected from the anterior vena cava of the Tibetan minipigs in the NC group, and after 3000 rpm centrifugation for 10 min, the supernatant was collected. All supernatants were stored in a refrigerator at −80 °C.

#### 2.2.1. Sudan IV Staining to Observe Lipid Deposition in the Aorta

After feeding the minipigs the HFC diet for 28 weeks, they were given excessive anesthesia and underwent bloodletting. The blood vessels from the aortic arch to the iliac part of the abdominal aorta were extracted, and the excess fat around the vessels was removed, and fixed in 10% neutral formaldehyde for 24 h. At a concentration of 10%, neutral formaldehyde can coagulate the protein and maintain the shape of the sample. The samples were then stained with Sudan IV dye (Sigma, Saint Louis, MO, USA) for 15 min, rinsed, and photographed. Image-Pro Plus 6.0 software was used to analyze lipid deposition in the whole aorta. Sudan IV dyes can bind with the lipids in the sample, making the lipids appear red.

#### 2.2.2. HE Staining to Observe the Pathological Morphology of Coronary Artery Tissue

After the animals were sacrificed, the coronary artery was isolated, immediately fixed in 10% formaldehyde, then dehydrated, transparent, embedded in wax, cut into 5 μm slices, patched, stained with HE dye (Thermo, Waltham, MA, USA), and finally mounted. Pathological sections of vascular tissue were scanned with a 2.0 RS Nana Zoomer digital slide scanner (Hamamatsu, Hamamatsu, Japan). NDP. view 2 software was used to measure the IMT (intima-media thickness).

#### 2.2.3. Blood Lipid Detection

After the minipigs were fed the HFC diet for 28 weeks, the pigs fasted for 16 h. Subsequently, the whole blood was collected and centrifuged at 3000 rpm for 10 min. The serum was separated to detect the levels of Glu, TC, TG, HDL-C, and LDL-C (Glu:glucose, TC: total cholesterol, TG: triacylglycerol, HDL-C: high-density lipoprotein, LDL-C: low-density lipoprotein) under the guidance of kit instructions (Medicalsystem Biotechnology Co., Ltd., Ningbo, China; lot numbers: 190926201, 191114101, 190422101, 190621201). The atherosclerosis index (AI) was calculated with the formula: AI = (TC − HDL-C)/HDL-C.

#### 2.2.4. Detection of Serum Inflammation, Oxidative Stress, and Vascular Endothelial Injury-Related Indicators

The commercially available kits of ox-LDL, CRP, TNF-α, IL-1β, and IL-6 (ox-LDL: oxidized low density lipoprotein, CRP: C-reactive protein, TNF-α: tumor necrosis factor-alpha, IL-1β: interleukin-1-beta, and IL-6: interleukin-6) were purchased from NanJing JianCheng Bioengineering Institute (Nanjing, China; lot number: all are 20210326), and vWF and ET-1 (ET-1: endothelin-1, vEF: ventricular ejection fraction) were bought from MultiSciences (Lianke) Biotech Co., Ltd. (Hangzhou, China). The corresponding indexes were detected using the ELISA method following the protocol.

### 2.3. Non-Targeted Metabolomics Analysis

All samples were melted at 4 °C. A total of 100 µL from each sample was placed in a 2 mL centrifuge tube. To each centrifuge tube was added 400 µL methanol (−20 °C), which was shaken for 60 s, and thoroughly mixed. After centrifugation at 12,000 rpm and 4 °C for 10 min, all of the supernatant was collected and transferred to a new 2 mL centrifuge tube for vacuum concentration and drying. A total of 150 µL 2-chlorophenylalanine (4 ppm) 80% methanol solution was redissolved, and the supernatant was filtered using a 0.22 µm membrane to obtain the samples to be tested. An amount of 20 µL was taken from each sample to be tested and mixed into QC samples (QC: quality control, used to correct the deviation of the analysis consequence of mixed samples and errors caused by the analysis instrument itself) and the remaining samples to be tested were used for LC-MS detection.

The chromatographic separation was accomplished in a Thermo Vanquish system equipped with an ACQUITY UPLC^®^ HSS T3 (150 × 2.1 mm, 1.8 µm, Waters, Milford, MA, USA) column maintained at 40 °C. The temperature of the autosampler was set at 8 °C. Gradient elution of analytes was carried out with 0.1% formic acid in water (A1) and 0.1% formic acid in acetonitrile (B1) or 5 mM ammonium formate in water (A2) and acetonitrile (B2) at a flow rate of 0.25 mL/min. An injection of 2 μL of each sample was analyzed after equilibration. An increasing linear gradient of solvent B (*v*/*v*) was used as follows: 0~1 min, 2% B2/B1; 1~9 min, 2%~50% B2/B1; 9~12 min, 50%~98% B2/B1; 12~13.5 min, 98% B2/B1; 13.5~14 min, 98%~2% B2/B1; 14~20 min, 2% B1-positive model (14~17 min, 2% B2-negative model). The ESI-MSn experiments were executed on the Thermo Q Exactive Plus mass spectrometer(Thermo, Waltham, MA, USA) with the spray voltage of 3.5 kV and −2.5 kV in positive and negative modes, respectively. Sheath gas and auxiliary gas were set at 30 and 10 arbitrary units, respectively. The capillary temperature was 325 °C. The analyzer scanned over a mass range of *m*/*z* 81–1000 for a full scan at a mass resolution of 70,000. Data-dependent acquisition (DDA) MS/MS experiments were performed with an HCD scan. The normalized collision energy was 30 eV.

Data preprocessing: Proteowizard software(v3.0.8789) was used to convert the original data obtained from the computer into mzXML format (xcms input file format); R’s XCMS package was used to filtrate peaks identification, peaks filtration, and peaks alignment. The data matrix including the mass-to-charge ratio (*m*/*z*), retention time, and intensity was obtained in order to establish metabolomics. The metabolites were identified as follows. Firstly, the metabolites were confirmed based on precise molecular weight (molecular weight error ≤ 30 ppm), and then the MS/MS fragments were analyzed using the Human Metabolome Database (HMDB) (http://www.hmdb.ca, accessed on 23 December 2020), METLIN (http://metlin.scripps.edu, accessed on 23 December 2020), Massbank (http://www.massbank.jp/, accessed on 23 December 2020), LipidMaps (http://www.lipidmaps.org, accessed on 23 December 2020), mzClound (https://www.mzcloud.org, accessed on 23 December 2020), and the self-established standard databases for annotation of metabolites.

### 2.4. Data Analysis

In untargeted metabolomics, we used the QC samples to evaluate the signal drift in the entire mass spectrometry data acquisition process. These drifts can be further identified, corrected, and used to improve the quality of the data using precise algorithms. In this process, the QC-RFSC algorithm from the R language statTarget package was used to correct the signal peaks of each feature (metabolite) in each sample, and the correction effect of each metabolite was recorded. The QC sample was a sample obtained by mixing equal amounts of all samples.

The standardization correction for metabolite was divided into three steps: 1. in-sample correction, that is, the abundance of all metabolites in the sample divided by the median metabolite abundance of the sample; 2. abundance matrix correction, that is, log conversion of all abundance values; 3. in-metabolite correction, that is, the abundance of all samples corresponding to the metabolite minus the average value of the metabolite abundance and then divided by the standard deviation of the metabolite abundance. Data were exported to Excel for subsequent analysis.

All charts were created using R language statistics, with $\bar{x}$ ± SEM used to show the differences between the two groups compared via Student’s *t*-test; *p* < 0.05 was statistically significant. The cor function in R language was used to calculate the correlation coefficient and the corPvalueStudent function was used to calculate the *p*-value.

## 3. Results

### 3.1. Identification of Differential Abundance Metabolites in the Development of AS Based on the Untargeted Metabolome

To investigate the metabolic characteristics of AS, Tibetan minipigs were used to build an AS model by feeding them with a high-fat diet. Blood was collected at 4, 8, 16, and 28 weeks in the AS group and at 16 and 28 weeks in the control groups which were fed a normal diet for metabolites identification (Figure 1A). Within four weeks of high-fat induction, the serum lipid level of Tibetan minipigs was significantly increased and hyperlipidemia gradually developed to form AS plaques at 16 weeks. Mature plaques were formed at 28 weeks. These were mainly distributed in the abdominal aorta and coronary artery, which is similar to the occurrence and development process and lesion sites of human clinical AS (Figure 1B). There were obvious AS lesions. Compared with the NC group, the Sudan IV staining of aortic vessels showed obvious lipid accumulation in the aortic arch of the AS group at 28 weeks (Figure 1B,C). Sudan IV can stain the lipids in blood vessels, and demonstrated lipid deposition and obvious AS lesions in blood vessels in the model group. Based on semi-quantitative analysis, the aortaventralis intra-media thickness and lipid accumulation area were significantly increased in the AS group compared with the NC group (Figure 1D,E and Appendix A). Moreover, a UMAP (Uniform Manifold Approximation and Projection) [16] analysis of metabolites in the AS group showed that samples from 4, 8, 16, and 28 weeks were clustered (Figure 1F).

### 3.2. Characteristics of Metabolites Enriched in the AS Model Group

To investigate the metabolites associated with AS, a hierarchical clustering analysis was performed using the metabolites in the NC and model group at 16 and 28 weeks after filtering with the adjusted *p*-value < 0.05 of the ANOVA. Four clusters were observed (Figure 2A). Based on heat map analysis, the metabolites in clusters 1 and 2 were associated with the changing of time, and the metabolites in clusters 3 and 4 were associated with AS. A total of 27 metabolites in cluster 3 were increased and 19 metabolites in cluster 4 were decreased in the model group compared with the NC group, at both 16 and 28 weeks (Figure 2A). Metabolite pathway enrichment analysis was performed using MetaboAnalyst [17]. Three pathways, i.e., sphingolipid metabolism, the pentose phosphate pathway, and glycine, serine, and threonine metabolism, were enriched in the AS cluster (Figure 2B). Increasing evidence has shown that sphingolipid metabolism is associated with AS by promoting inflammatory responses, cholesterol efflux, and aggregation of particles in the aorta [18,19,20]. In this study, the three sphingolipid metabolism-related metabolites, i.e., sphinganine 1-phosphate, sphingosine 1-phosphate, and sphingosine, were enriched in the AS cluster (Figure 2C). The associations between sphingolipid metabolism and the AS index, IMT, endothelial injury markers, and inflammatory markers were analyzed (Appendix A). Consequently, significant positive correlations between sphingolipid metabolism and AS index, endothelial injury, and inflammation were observed (Figure 2D and Appendix A). The relative abundance of metabolites in all the samples can be seen in Appendix A.

In addition, several products of glucose oxidation were identified in the AS cluster, including glucuronic acid, 5-keto-d-gluconate, gluconolactone, and gluconic acid (Figure 3A). These products exhibited a significant positive correlation with the AS index, IMT, endothelial injury (ET-1 and vWF), and inflammation (CRP and IL-6) (Figure 3B and Appendix A). There was no significant difference between blood Glu and D-glucose (Appendix A).

### 3.3. Phenylalanine Metabolism Was Reduced in the AS Model Group

Based on the enrichment analysis, phenylalanine metabolism was reduced in the AS group (Figure 4A). We analyzed the correlations between the AS index and phenylalanine metabolism-related metabolites. Negative correlations between the AS index and phenylalanine metabolism-related metabolites such as L-Phenylalanine, Hydroxyphenylacetic acid, Phenylacetylglycine, Phenylpyruvic acid, and 2-Phenylacetamide were observed (Figure 4B,C). Then, we performed a correlation analysis between these metabolites and IMT, endothelial injury markers (ET-1, vEF), and inflammatory markers (CRP, TNFA, IL-1B, and IL-6). Consequently, Phenylacetylglycine, Phenylpyruvic acid, and 2-Phenylacetamide were negatively correlated with ET-1, and Hydroxyphenylacetic acid, Phenylpyruvic acid, and 2-Phenylacetamide were negatively correlated with vWF (Figure 4C and Appendix A). This may be because phenylalanine metabolism-related metabolites are associated with vascular injury in Tibetan miniature pigs.

### 3.4. Gluconolactone Was Enriched in the Late Stage of AS

In a previous analysis, we found that the metabolites in cluster 3 were enriched in the model group. To determine whether these metabolites gradually increase in the development of AS, the abundance of metabolites in cluster 3 in the model group at 4, 8, 16, and 28 weeks and in the NC group at 16 and 28 weeks were compared. We found that with the increase in time, gluconolactone gradually increased in the model group, and a lower level of gluconolactone was observed in the NC group at 16 and 28 weeks (Figure 5A). The increase in gluconolactone could be due to glucose oxidation by glucose oxidase and lead to the production of FADH2 from FAD [21]. In a previous analysis, we found that gluconolactone and gluconic acid were associated with the AS index, IMT, endothelial injury, and inflammation (Figure 3B and Appendix A). In particular, a higher significantly positive correlation between gluconolactone and the AS index (R = 0.74, *p* = 8.4 × 10^−5^) (Appendix A) and a higher significantly positive correlation between gluconolactone and vascular injury marker vWF (R = 0.9, *p* < 2.2 × 10^−6^) were observed (Figure 5B). This may mean that gluconolactone is associated with vascular injury and AS. Increasing evidence has shown that gluconolactone is associated with the dysfunction of cardiovascular disease [22]. Gluconolactone and gluconic acid are part of the PPP (pentose phosphate pathway). However, in this study, other metabolites such as glucose, 6-phosphogluconic acid, D-ribose, and ribose 1,5-bisphosphate in the PPP were not correlated with the AS index (Figure 5C and Appendix A). Taken together, the above observations suggest that gradually enriched gluconolactone could provide clues in the development of AS.

### 3.5. Biopterin Was Reduced in the Late Stage of AS

For the metabolites reduced in cluster 4 of the model group, we observed a gradual decrease in certain metabolites such as biopterin, L-Homophenylalanine, and Tranylcypromine as the development of AS progressed from 4 weeks to 28 weeks. In contrast, these metabolites maintained a higher level in the NC group at 16 and 28 weeks, as determined by the same comparison method used above (Figure 5D and Appendix A).

Biopterin is an oxidative product of BH4 that can reduce coronary artery disease and improve endothelial function by decreasing NO (nitric oxide) production [23]. Negative correlations between the AS index and biopterin (R = −0.63, *p* = 0.013) were observed (Figure 5E and Appendix A). In addition, biopterin negatively correlated with EF-1 (R = −0.63, *p* = 0.037) (Appendix A). This suggests that vascular injury and AS are closely related to levels of biopterin.

## 4. Discussion

The Tibetan minipig is a characteristic breed of China that has been widely used in cardiovascular disease and diabetes research. Our previous studies showed that Tibetan minipigs are susceptible to the formation of AS lesions due to high-fat diets, and obvious lipid disorders and inflammatory reactions can be observed [13]. The Tibetan minipig has a clear advantage as an AS model animal for studying the pathogenesis of human AS. The AS model of Tibetan minipigs is helpful for explaining the metabolic mechanism of AS induced by a high-fat and high-cholesterol diet from the perspective of metabolomics. However, the pathological characteristics and mechanism of the Tibetan minipig AS model have not been fully clarified. In this study, we built an AS animal model using Tibetan minipigs fed with a high-fat and high-cholesterol diet to identify the changes in serum metabolites at 4, 8, 16, and 28 weeks and find related metabolic pathways. Based on the untargeted metabolome, the differential abundance of metabolites in the development of AS was determined. This implies that AS lesions were gradually formed from the early stage to the late stage through feeding with a high-fat diet. Another piece of evidence showed that 3-hydroxyphenyl acetic acid can reduce blood pressure and vascular injury through the release of nitric oxide [24]. Taken together, we speculated that phenylalanine metabolism dysfunction could influence vascular injury and AS through the release of nitric oxide in the Tibetan minipigs. We found that sphingolipid metabolism and glucose oxidation were enriched in the AS group and phenylalanine metabolism was reduced in the AS group.

Sphingolipids include sphingomyelins, glucosylceramides, and sphingosine. These sphingolipids are localized in lipid bilayers, and ceramides are essential precursors of most of the complex sphingolipids [25]. Clinical studies with large cohorts have shown that serum ceramide levels are strong predictors of coronary artery disease. Serum ceramide levels also predict atherosclerotic plaque instability and detrimental outcomes of coronary artery disease, including death. Ceramides also accumulate in atherosclerotic plaques, where they have been implicated in the onset of lipoprotein aggregation [26]. In mice, the inhibition of ceramide biosynthesis ameliorates hallmark features of cardiometabolic disease, including insulin resistance, glucose intolerance, diabetes, and atherosclerotic plaque formation [27]. In this study, sphingomyelins, sphingosine, and galactosylsphingosine were enriched during the occurrence of AS in Tibetan miniature pigs.

Atherosclerosis is a chronic inflammatory disease with a complex pathological mechanism that involves the imbalance of lipid metabolism, the oxidative stress response, and the inflammatory response of vascular endothelial cells. The extent to which vascular endothelial cells are injured heavily influences the occurrence and progression of AS [28]. In the development of AS, reducing biopterin is significantly associated with the aggravation of endothelial injury. It has been reported that BH4 (tetrahydrobiopterin), a biopterin reduction product, can reduce coronary artery disease and improve endothelial function by decreasing NO production [23,29]. This further proves that biopterin was closely associated with the development of AS. In the development of AS, gluconolactone is more highly significantly positively correlated with endothelial injury factor vWF. During endothelial injury, the glucose is oxidized, which leads to the accumulation of gluconolactone. Increasing gluconolactone could indicate the redox state in the body which is associated with a series of physiological and metabolic effects for cardiovascular and cerebrovascular diseases. In atherosclerosis, excess reactive oxygen species (ROS) are generated to induce the oxidative stress which is implicated in vascular injury and inflammation [30]. Research has shown that glucose oxidation can induce oxidative stress [31]. In this study, several products of glucose oxidation were increased during the occurrence of AS in Tibetan miniature pigs, including gluconolactone, gluconic acid, 5-keto-d-gluconate, and glucuronic acid (Figure 3A). These metabolites are involved in two glucose metabolic pathways, one of which is gluconic acid metabolism. This is a classical glucose oxidation process that does not produce ATP. The gluconolactone could be due to glucose oxidation by glucose oxidase and the gluconic acid could be due to the hydrolysis of gluconolactone catalyzed by lactonase or spontaneously. Gluconic acid can be converted to two 5-keto-D-gluconates (5KGA) and 2-keto-D-gluconate (2KGA) [32]. In glucose oxidation, a large amount of hydrogen peroxide (H_2_O_2_) can be created, which can promote oxidative stress [33]. The other pathway is the glucuronate pathway. Glucuronic acid can be converted from UDP-glucuronic acid which is oxidized from UDP-glucose by NAD+ and UDP-glucose dehydrogenase.

In the development of AS in Tibetan miniature pigs, gluconolactone is enriched in the late stage of AS whereas biopterin is enriched in the early stage of AS in Tibetan miniature pigs. Gluconolactone and biopterin could provide clues for indicating the development of AS in Tibetan miniature pigs. This study could provide a basis for the study of atherosclerosis. However, many diseases in humans have complex etiology and pathogenesis, while animal models can only simulate a part of them. The occurrence and development of atherosclerosis is multi-factor and multi-step, and is the result of genetic factors and environmental factors. Even when different individuals are affected by the same factors, the results are different. Therefore, animal models may not fully reflect the complexity of human diseases, thereby limiting the in-depth understanding of diseases and the development of treatment methods. The findings of this study need further verification in animal and clinical research.

## Figures and Tables

**Figure 1 nutrients-15-04425-f001:**
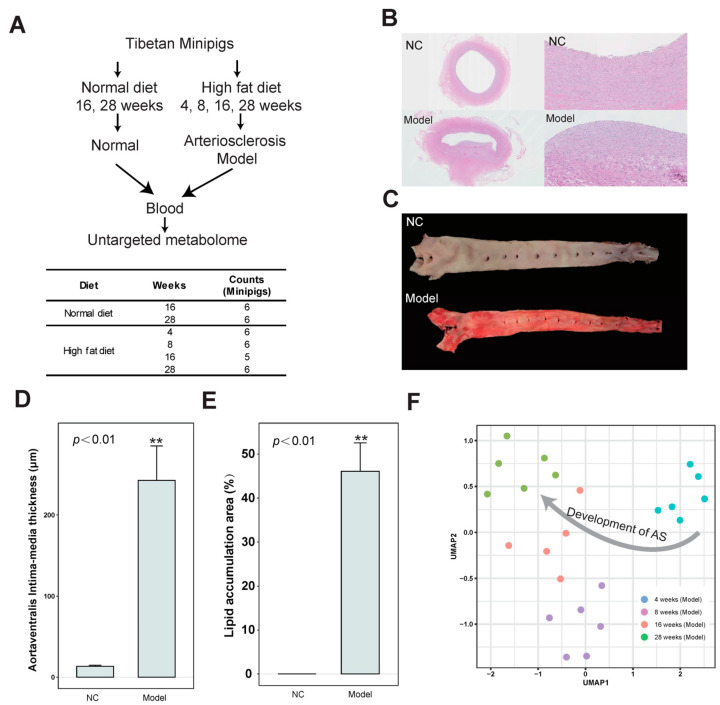
Identification of differential abundance of metabolites in the development of AS based on the untargeted metabolome in Tibetan miniature pig serum. (**A**) Workflow of identification of metabolites of serum in the development of AS. (**B**) Comparison of HE staining of abdominal aortic vessels between model and NC groups. (**C**) Comparison of Sudan IV staining of aortic vessels between model and NC groups. (**D**) Bar graph showing the difference in aortaventralis intima-media thickness between the NC and AS groups. The error bar indicates the standard error of the mean (SEM). (**E**) Bar graph showing difference in lipid accumulation area (%) between the NC and AS groups. The error bar indicates the standard error of the mean (SEM). (**F**) UMAP (Uniform Manifold Approximation and Projection) visualization for the model group at 4, 8, 16, and 28 weeks. ** indicates *p* < 0.01.

**Figure 2 nutrients-15-04425-f002:**
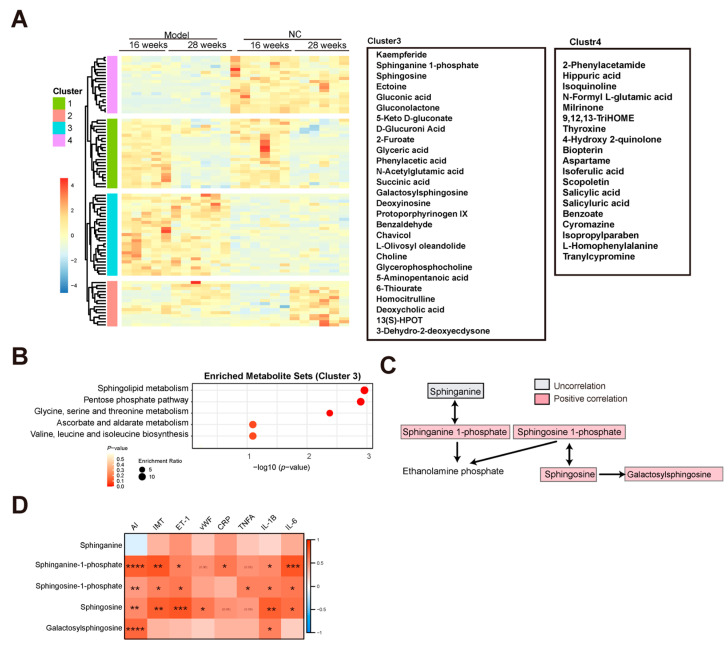
Characteristics of metabolites enriched in the AS model group. (**A**) Heat map showing hierarchical clustering for NC and AS groups at 16 and 28 weeks. (**B**) Enriched metabolite sets from the KEGG database for AS model group (cluster 3). (**C**) Flow chart showing the sphingolipid metabolism. Red and blue indicate positive and negative correlations between AS and metabolites, respectively. Gray indicates no correlation between AS and metabolites. (**D**) Correlation analysis between sphingolipid metabolism and AS index, IMT, endothelial injury, and inflammation; * indicates *p* < 0.05; ** indicates *p* < 0.01; *** indicates *p* < 0.001; **** indicates *p* < 0.0001.

**Figure 3 nutrients-15-04425-f003:**
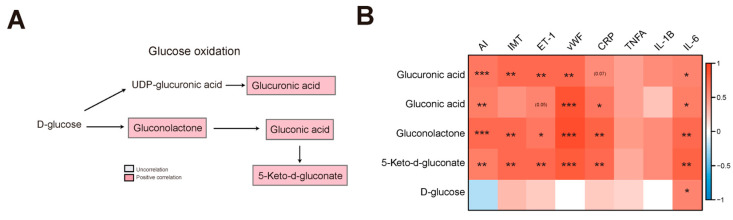
Glucose oxidation enriched in the AS model group. (**A**) Flow chart showing the glucose oxidation. Red and blue indicate positive and negative correlations between AS and metabolites, respectively. Gray indicates no correlation between AS and metabolites. (**B**) Correlation analysis between glucose oxidation and AS index, IMT, endothelial injury, and inflammation; * indicates *p* < 0.05; ** indicates *p* < 0.01; *** indicates *p* < 0.001.

**Figure 4 nutrients-15-04425-f004:**
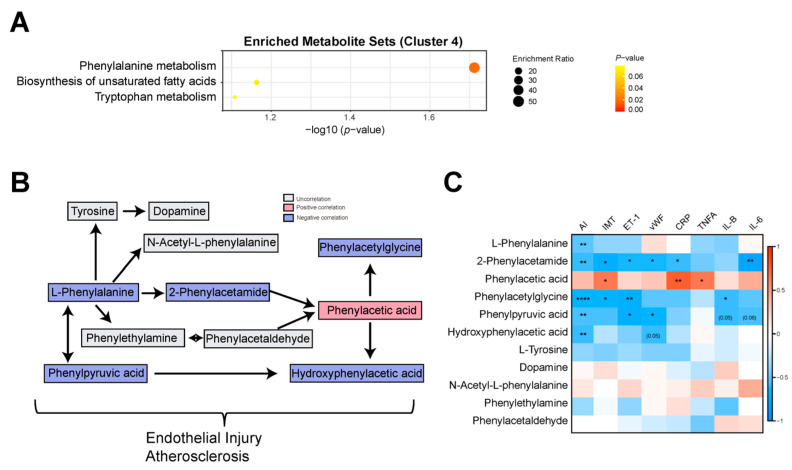
Phenylalanine metabolism was reduced in the AS model group. (**A**) Enriched metabolite sets from the KEGG database for the NC group (cluster 4). (**B**) Flow chart showing L-phenylalanine metabolism. Red and blue indicate positive and negative correlations between AS and metabolites, respectively. Gray indicates no correlation between AS and metabolites. (**C**) Correlation analysis between phenylalanine metabolism and AS index, IMT, endothelial injury, and inflammation; * indicates *p* < 0.05; ** indicates *p* < 0.01; **** indicates *p* < 0.0001.

**Figure 5 nutrients-15-04425-f005:**
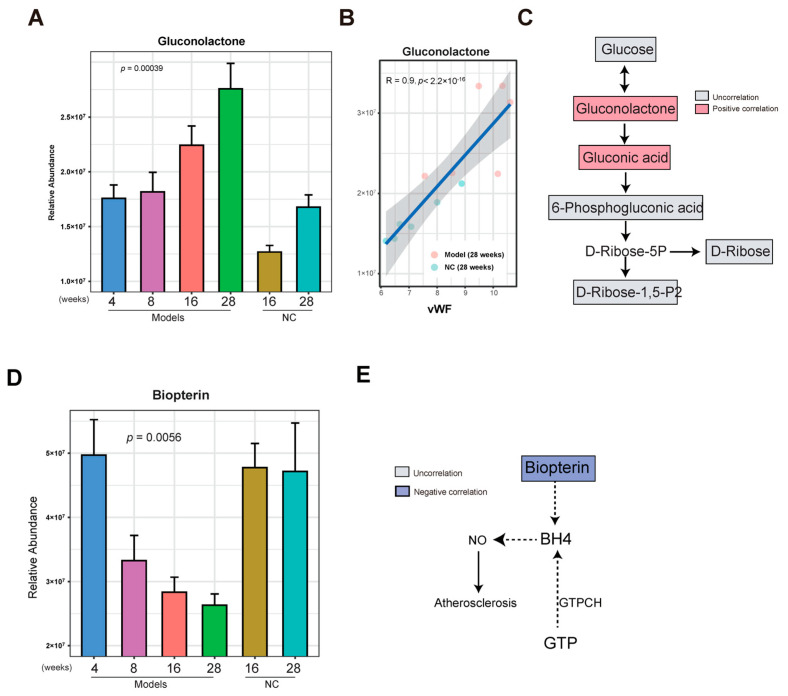
Gluconolactone was enriched and biopterin was reduced in the late stage of AS. (**A**) Bar graph showing the abundance of gluconolactone in the AS model group at 4,8,16 and 28 weeks and the NC group at 16 and 28 weeks. The error bar indicates the standard error of the mean (SEM). The *p*-value was calculated by ANOVA analysis. (**B**) Correlation analysis between gluconolactone and vEF. (**C**) Flow chart showing the pentose phosphate pathway. Red and blue indicate positive and negative correlations between AS and metabolites, respectively. Gray indicates no correlation between AS and metabolites. (**D**) Bar graph showing the abundance of biopterin in the AS model group at 4,8,16 and 28 weeks and in the NC group at 16 and 28 weeks. The error bar indicates the standard error of the mean (SEM). The *p*-value was calculated by ANOVA analysis. (**E**) Flow chart showing the pentose phosphate pathway. Red and blue indicate positive and negative correlations between AS and metabolites, respectively. Gray indicates no correlation between AS and metabolites.

## Data Availability

The dataset supporting the conclusions of this article is included within the article.

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
