# Peer review of "Characteristics of Metabolites in the Development of Atherosclerosis in Tibetan Minipigs Determined Using Untargeted Metabolomics"

_nutrients, 2023, doi:10.3390/nu15204425_

Round 1

Reviewer 1 Report

General Remarks:

1.    The manuscript's overall language requires revision for grammar and tense to enhance readability.

2.    The language should be tailored to scientific readers rather than a general audience.

Introduction:

1.    Provide a more elaborate explanation of the concept of "glucose oxidation," particularly its common denotation is the breakdown of glucose for ATP generation.

2.    Refine the title to be more indicative and specific, mentioning the animal model used.

3.    While the paper emphasizes human model similarity, no experiments were conducted to support this claim. Consider adding an additional experiment involving human blood profiling.

4.    Ensure proper citation inclusion (37-39).

5.    Revise sentences (41-44) for conciseness, clarity, and scientific language.

6.    Organize enumerated points for conciseness (45-47).

7.    Revise tense usage (51) accordingly.

8.    Enhance assertiveness when describing phenomena (52).

9.    Provide more specificity in describing biological processes (54), including the inflammatory cascade involved.

10. Rephrasing to be considered (60)

11. References not in English (69, 72, 73)

Materials and Methods:

1.    Clarify how the selection of pigs was carried out. Were they from the same litter?

2.    Explain the process of assigning pigs to each group and specify the number in each group.

3.    Provide detailed information about the food administered to the NC group.

4.    Clarify the reference of "They" in context (91).

5.    Is the "3000 rpm" for a centrifuge or other equipment.

6.    Why are the NC group measurement points not the same as the AS model (only two vs. four).

7.    Elaborate on the rationale for reserving and fixing excess fat in 10% neutral formaldehyde.

8.    Clearly define the nature of the sample; is it blood? Specify the number of experimental replicates used and describe how measurements were normalized.

9.    Clarify the quantification of metabolites, including the method used (e.g., post-column infused internal standard - PCI-IS).

10. Specify the LC-MS instrument used and its settings; consider adding this information to supplementary material if adjustments were manual.

11. Detail how metabolites were identified, including the database or library used.

12. Mention any data filtration or curation, were data tested for normality that influenced the selection of the student-t test? What are the names of the packages used in R?

13. Provide the names of R packages used.

14. Several methods were mentioned in the results section, but not highlighted in the methods section, like the purpose of using Sudan IV staining, and the reason for selecting UMAP in particular for nonlinear dimensionality reduction

Results:

1.    What is the result of Sudan IV supposed to convey? Not clear

2.    The word resultantly is used 5 times throughout the manuscript, maybe consider more lexical diversity, and only when needed.

3.    Define acronyms such as "vWF" (line 208).

4.    The reference is a bit outdated (line 262), is there any current research supporting this? Elaborate on the utility of the point and provide additional evidence.

5.    Discuss any relevant literature concerning sphingolipids and their impact on AS-related systemic inflammation.

6.    Address limitations encountered in the experiment.

Author Response

Dear Editor and reviewers,

Thank you for your letter dated August 28. We were pleased to know that our work was rated as potentially acceptable for publication in Journal, subject to adequate revision. We thank the reviewers for the time and effort that they have put into reviewing the previous version of the manuscript. Their suggestions have enabled us to improve our work. Based on the instructions provided in your letter, we uploaded the file of the revised manuscript. Accordingly, we have uploaded a copy of the original manuscript with all the changes highlighted by using the track changes mode in Word.

Appended to this letter is our point-by-point response to the comments raised by the reviewers. The comments are reproduced and our responses are given directly afterward.

We would like also to thank you for allowing us to resubmit a revised copy of the manuscript.

We hope that the revised manuscript is accepted for publication in the Nutrients.

Reviewer 1

Comments to the Author

Here are the general comments from the reviewer.

We thank the reviewer for the very interesting comments. In fact “The manuscript's overall language requires revision for grammar and tense to enhance readability”.Regarding the suggestion about the language, we changed  and modified it.

Another issue on the language section is that ”The language should be tailored to scientific readers rather than a general audience.” We are grateful for the suggestion. To be more clear and in accordance with the reviewer concerns, We called in a professional to modify manuscript.

Minor comments:

1.Provide a more elaborate explanation of the concept of "glucose oxidation," particularly its common denotation is the breakdown of glucose for ATP generation.

Reply: In this study, several products of glucose oxidation were increased in the AS group, including gluconolactone, gluconic acid, 5-keto-d-gluconate, and glucuronic acid (Fig. 3A).These metabolites are involved in two glucose metabolic pathways, one of which is gluconic acid metabolism. This is a classical glucose oxidation process that does not produce ATP. The gluconolactone could be from glucose oxidation by glucose oxidase and the gluconic acid can be from hydrolysis of gluconolactone catalyzed by lactonase or spontaneously. The gluconic acid can be converted to two 5-keto-D-gluconates (5KGA) and 2-keto-D-gluconate (2KGA) (Toyama et al., 2007). The other pathway is the glucuronate pathway. The glucuronic acid can be converted from UDP-glucuronic acid which is oxidized from UDP-glucose by NAD+ and UDP-glucose dehydrogenase. We discussed this issue in discussion.(line375-386)

2. Refine the title to be more indicative and specific, mentioning the animal model used.

Reply: Thank you for  the suggestion. The revised version of manuscript, the title has been changed to “Characteristics of metabolites in the development of Atherosclerosis in Tibetan minipigs determined using untargeted metabolomics”.(line2-4)

3. While the paper emphasizes human model similarity, no experiments were conducted to support this claim. Consider adding an additional experiment involving human blood profiling

.

Reply: Thank you for your valuable suggestions. We discussed this limitation in this paper, the human blood analysis involves ethical approval, we cannot integrate it with clinical practice at present study. If there is an opportunity later, we also hope to cooperate with clinical practice. Actually, in present study, this AS model have the very similar etiological and pathological characteristics of human AS. This AS model of central obesity is accompanied by insulin resistance and hypertension characterized by chronic inflammation and lipid metabolism disorder, which is very similar to the etiological and pathological characteristics of human AS formation(Pan Y et al., 2019, 陈民利,2018). Our research group found in the previous study that Tibetan miniature pigs as a model of atherosclerosis were similar to human atherosclerosis, and the Chinese literature as auxiliary reference was deleted in article.

陈民利.西藏小型猪和五指山小型猪的生理特点及其动脉粥样硬化的发病特征比较研究[J].实验动物与比较医学,2018,38(05):325-328.

4. Ensure proper citation inclusion (37-39).

Reply: We have added a new Reference.(line 46)

5.Revise sentences (41-44) for conciseness, clarity, and scientific language.

Reply: The sentences “Long-term consumption of a diet inched in high fat, high cholesterol, and high sugar will cause lipid metabolism disorders, increase the risk of AS and promote plaque development in patients with AS.” had been changed to ” Long-term consumption of the  high fat, high cholesterol, and high sugar diet will increase the risk of AS”.(line 49-51)

6. Organize enumerated points for conciseness (45-47).

Reply: “The occurrence and development of AS are involved in the accumulation of vascular lipids, activation of the immune system, inflammation, oxidative stress and oxidized low-density lipoprotein, activation of endothelial cells, the proliferation of arterial smooth muscle cells and activation of macrophages, and formation of foam cells. ” had been changed to ”The occurrence and development of AS are involved in the accumulation of vascular lipids, inflammation, oxidative stress, etc. ”(line52-55)

7. Revise tense usage (51) accordingly.

Reply: “The low-density lipoprotein (LDL) was oxidized to oxLDL by low-density lipoprotein receptors (LDLR).” had been changed to ” The low-density lipoprotein (LDL) is oxidized to oxLDL by low-density lipoprotein receptors (LDLR).”(line 58-60)

8. Enhance assertiveness when describing phenomena (52).

Reply: “The ox-LDL may bind to endothelial cell surface lectin-like oxidized low-density lipoprotein receptor 1 (LOX1)…”changed” Ox-LDL binds to endothelial cell surface lectin-like oxidized low-density lipoprotein receptor 1 (LOX1)…”(line 59-63)

9. Provide more specificity in describing biological processes (54), including the inflammatory cascade involved.

Reply: “…to enter endothelial cells and activate endothelial cells to produce an inflammatory cascade…”changed” …to enter endothelial cells and activate endothelial cells to produce an inflammatory cascade, such as toll-like receptor pathway and arachidonic acid pathway activation…”(line61-63)

10. Rephrasing to be considered (60)

Reply: “However, these small molecule metabolites have the potential to become biomarkers of cardiovascular diseases and the metabolic mechanism of AS has not been fully understood.” changed ”These small molecule metabolites have the potential to become biomarkers of cardiovascular disease and provide a new direction for the prevention and treatment of atherosclerosis.“(line69-72)

11. References not in English (69, 72, 73)

Reply: We had modified references.(line 80,83)

Materials and Methods:

1.Clarify how the selection of pigs was carried out. Were they from the same litter?

Reply: A total of 12 miniature pigs were selected based on a combination of sex, weight and age, regardless of whether they were in the same litter.Conventional male Tibetan minipigs, aged 4-5 months, weighing 8-12 kg, 12 pigs (line90-91)

2. Explain the process of assigning pigs to each group and specify the number in each group.

Reply: After 1 month of adaptive feeding in the laboratory, 6 animals were randomly assigned to each group according to body weight and blood biochemical indexes, and there was no significant difference between the two groups. We have added it in the methods. (line97-100)

3. Provide detailed information about the food administered to the NC group.

Reply: The NC group were fed with 100% basal diet. We have added it in methods.(line 108-109)

4. Clarify the reference of "They" in context (91).

Reply: The "They" had been changed to AS model group”. The text has been modified. (line 106)

5. Is the "3000 rpm" for a centrifuge or other equipment.

Reply: "3000 rpm" is "3000 rpm centrifugation". The text has been added. (line 112)

6. Why are the NC group measurement points not the same as the AS model (only two vs. four).

Reply: In this present study, we focus on the changes of metabolites at the different stages in the model group. On the other hand, we don't have enough minipigs are feed due to insufficient funding support.

7. Elaborate on the rationale for reserving and fixing excess fat in 10% neutral formaldehyde.

Reply: We have added “At 10% formaldehyde can coagulate the protein and maintain the shape of the sample, and Sudan IV dyes can bind with the lipids in the sample, the lipids will show red.”(line117-119)

8. Clearly define the nature of the sample; is it blood? Specify the number of experimental replicates used and describe how measurements were normalized.

Reply: In this study, the sample is serum, not is blood. Each group had 6 samples per stage. The standardization correction for metabolite is divided into three steps: 1. In-sample correction, that is, the abundance of all metabolites in the sample divided by the median metabolite abundance of the sample; 2. Abundance matrix correction, that is, log conversion of all abundance values; 3. In-metabolite correction, that is, the abundance of all samples corresponding to the metabolite minus the average value of the metabolite abundance and then divided by the standard deviation of the metabolite abundance, and exported the data to excel for subsequent analysis. We had added it in methods. (line140-152)

9. Clarify the quantification of metabolites, including the method used (e.g., post-column infused internal standard - PCI-IS).

Reply: The relative quantification of non-target metabolites was calculated by peak area. Data preprocessing: Proteowizard software was used to convert the original data obtained from the computer into mzXML format (xcms input file format); R's XCMS package was used to filtrate peaks identification, peaks filtration and peaks alignment. The data matrix including the mass to charge ratio (m/z), retention time and intensity was obtained in order to establish metabolomics. We have added it in methods. (line138-142)

10. Specify the LC-MS instrument used and its settings; consider adding this information to supplementary material if adjustments were manual.

Reply: The chromatographic separation was accomplished in an Thermo vanquish system equipped with an ACQUITY UPLC® HSS T3 (150 × 2.1 mm, 1.8 µm, Waters) column maintained at 40 . The temperature of the autosampler was set 8 . Gradient elution of analytes was carried out with 0.1% formic acid in water (A1) and 0.1% formic acid in acetonitrile (B1) or 5 mM ammonium formate in water (A2) and acetonitrile (B2) at a flow rate of 0.25 mL/min. Injection of 2 μL of each sample was analysed after equilibration. An increasing linear gradient of solvent B (v/v) was used as follows: 0~1 min, 2% B2/B1; 1~9 min, 2%~50% B2/B1; 9~12 min, 50%~98% B2/B1; 12~13.5 min, 98% B2/B1; 13.5~14 min, 98%~2% B2/B1; 14~20 min, 2% B1- positive model (14~17 min, 2% B2-negative model). The ESI-MSn experiments were executed on the Thermo Q Exactive Plus mass spectrometer with the spray voltage of 3.5 kV and -2.5 kV in positive and negative modes, respectively. Sheath gas and auxiliary gas were set at 30 and 10 arbitrary units, respectively. The capillary temperature was 325 . The analyzer scanned over a mass range of m/z 81-1 000 for full scan at a mass resolution of 70 000. Data dependent acquisition (DDA) MS/MS experiments were performed with HCD scan. The normalized collision energy was 30 eV. We added it in material. (line132-146)

11. Detail how metabolites were identified, including the database or library used.

Reply: The of metabolites identified: firstly, the metabolites were confirmed based on precise molecular weight (molecular weight error <= 30ppm), and then the MS/MS fragment were analyzed with Human Metabolome Database (HMDB) (http://www.hmdb.ca),  METLIN (http://metlin.scripps.edu), Massbank (http://www.massbank.jp/), LipidMaps (http://www.lipidmaps.org), mzClound(https://www.mzcloud.org), and the self-established standards database by BGI genomics for annotation of metabolites. We added it in material.(line146-152)

12. Mention any data filtration or curation, were data tested for normality that influenced the selection of the student-t test? What are the names of the packages used in R?

13. Provide the names of R packages used.

Reply 12 and 13: In the method, we provide detailed procedures for data generation, filtering, correction and names of the R packages.The MetaboAnalyst comprehensive platform was used for pathway enrichment Analysis ( https://www.metaboanalyst.ca/). The pheatmap, ggplot2 and ggpubr packages were used for data visualization.The uwot package was used for UMAP (Uniform Manifold Approximation and Projection) analysis.

14. Several methods were mentioned in the results section, but not highlighted in the methods section, like the purpose of using Sudan IV staining, and the reason for selecting UMAP in particular for nonlinear dimensionality reduction

Reply: The Sudan IV staining which stains lipids in blood vessels, demonstrated lipid deposition in blood vessels in the model group.UMAP is a nonlinear dimensionality reduction algorithm that keeps the local and global features of the data by building local structures between data samples. This algorithm could better show the development stages of things. For example, in the single-cell data transcriptome analysis, UMAP is used to analyze the developmental trajectory of cells. In the present study, the UMAP analysis showed that AS gradually formed from the early stage to the late stage by feeding a high-fat diet. We added it in methods and result. (line117-119,183-188)

Results:

1. What is the result of Sudan IV supposed to convey? Not clear

Reply: The  Sudan IV can stain the lipids in blood vessels, that demonstrated lipid deposition and obvious AS lesions in blood vessels in model group. (line 176-178)

2. The word resultantly is used 5 times throughout the manuscript, maybe consider more lexical diversity, and only when needed.

Reply: We replaced several of the ‘consequence’ with ‘result.’ (line 209, 261)

3. Define acronyms such as "vWF" (line 208).

Reply: We have added the vWF definition . (line218-221)

4. The reference is a bit outdated (line 262), is there any current research supporting this? Elaborate on the utility of the point and provide additional evidence.

Reply: We have added another reference. (line 362)

5. Discuss any relevant literature concerning sphingolipids and their impact on AS-related systemic inflammation.

Reply: sphingolipids, including sphingomyelins, glucosylceramides and sphingosine. These sphingolipids are localized in lipid bilayers, and Ceramides are essential precursors of most of the complex sphingolipids. (Choi et al., 2021) The clinical studies with large cohorts reveal that serum ceramide levels are strong predictors of coronary artery disease. Serum ceramide levels also predict atherosclerotic plaque instability and detrimental outcomes of coronary artery disease, including death. Ceramides is also accumulated in atherosclerotic plaques, where they have been implicated in the onset of lipoprotein aggregation(Laaksonen et al., 2016) In mice, inhibition of ceramide biosynthesis ameliorates hallmark features of cardiometabolic disease, including insulin resistance, glucose intolerance, diabetes, atherosclerotic plaque formation(Park et al., 2008). And we have add it in discuss. (line345-354)

6. Address limitations encountered in the experiment.

Reply: We encountered many difficulties in the experiment. First of all, the individual differences among miniature pigs were large, and the data were not uniform, which was quite different from the expected experimental results. As a solution, we abandoned more metabolites with large differences and left those with small differences for analysis. On the other hand, we also encountered the problem of insufficient funds, so we chose experimental samples and discarded some samples

Reviewer 2 Report

In this manuscript, the authors studied atherosclerosis. Atherosclerosis (AS) is a progressive disease linked to lipid metabolism disorders. Authors established an AS model in Tibetan minipigs using a high-fat diet. Through untargeted metabolomics, they found increased sphingolipid metabolism and glucose oxidation and reduced phenylalanine metabolism in AS. Gluconolactone and biopterin emerged as potential AS development biomarkers, offering fresh insights into AS metabolism.

While the manuscript is well-structured and effectively communicates results, certain critical aspects require further clarification.

1. Please elaborate untargeted metabolomics approach.

2. How the metabolite annotation is validated? Whether authors prepare a pooled sample to create a targeted list?

3. How many internal standards were spiked in the extraction solvent to normalize the analytes responses in the samples as different molecules ionize differently in the mass spectrometry source?

3. Please clearly state in the abstract results that there was an observed change in lipometabolite levels between the control and AS groups in plasma.

4. I don’t think oxidative stress could contribute to glucose oxidation. Please provide a reference.

5. The results require a more thorough discussion, substantiated by relevant published materials.

Author Response

Dear Editor and reviewers,

Thank you for your letter dated August 28. We were pleased to know that our work was rated as potentially acceptable for publication in Journal, subject to adequate revision. We thank the reviewers for the time and effort that they have put into reviewing the previous version of the manuscript. Their suggestions have enabled us to improve our work. Based on the instructions provided in your letter, we uploaded the file of the revised manuscript. Accordingly, we have uploaded a copy of the original manuscript with all the changes highlighted by using the track changes mode in Word.

Appended to this letter is our point-by-point response to the comments raised by the reviewers. The comments are reproduced and our responses are given directly afterward.

We would like also to thank you for allowing us to resubmit a revised copy of the manuscript.

We hope that the revised manuscript is accepted for publication in the Nutrients.

Reviewer 2

  1. Please elaborate untargeted metabolomics approach.

Reply: We were added metabolomics approach in materials and methods. (line132-152)

  1. How the metabolite annotation is validated? Whether authors prepare a pooled sample to create a targeted list?

Reply: The identification of metabolites was first confirmed based on precise molecular weight (molecular weight error <= 30ppm), and then the MS/MS fragment analyzed by Human Metabolome Database (HMDB) (http://www.hmdb.ca), model METLIN (http://metlin.scripps.edu), Massbank (http://www.massbank.jp/), LipidMaps (http://www.lipidmaps.org), mzClound(https://www.mzcloud.org), and the self-established standards database. We were added it in material and methods. (line146-152)

  1. How many internal standards were spiked in the extraction solvent to normalize the analytes responses in the samples as different molecules ionize differently in the mass spectrometry source?

In this untargeted metabolomics study, we did not use the internal standards for data normalization. We used the QC (quality control) samples for evaluate the signal drift in the entire mass spectrometry data acquisition process. These drifts can be further identified, corrected, and improve the quality of the data using precise algorithms. In this process, the QC-RFSC algorithm from the R language statTarget package is used to correct the signal peaks of each feature (metabolite) in each sample, and the correction effect of each metabolite is recorded. The QC sample is a sample obtained by mixing equal amounts of all samples. We were added it in material and methods. (line132-139)

  1. Please clearly state in the abstract results that there was an observed change in lipometabolite levels between the control and AS groups in plasma.

Reply: regarding to lipometabolite, we found that based on the enrichment analysis, the sphingolipid metabolism was enriched in AS group (Fig 2C). In present study, the three sphingolipid metabolism-related metabolites including sphinganine 1-phosphate, sphingosine 1-phosphate, and sphingosine were enriched in the AS cluster from cluster analysis (Fig 2B).

  1. I don’t think oxidative stress could contribute to glucose oxidation. Please provide a reference.

Reply: We changed the statement “the glucose oxidation could be associated with oxidative stress” Actually, increasing evidences show that glucose oxidation could induce oxidative stress (Liu et al., 2012). In this study, several products of glucose oxidation were increased in the AS group, including gluconolactone, gluconic acid, 5-keto-d-gluconate, and glucuronic acid (Fig. 3A). The enriched gluconolactone could be from glucose oxidation by glucose oxidase and lead to produce FADH2 from FAD and the gluconic acid can be from hydrolysis of gluconolactone catalyzed by lactonase or spontaneously. The gluconic acid can be converted to two 5-keto-D-gluconates (5KGA) and 2-keto-D-gluconate (2KGA) (Toyama et al., 2007). In glucose oxidation, A large amount of hydrogen peroxide (H2O2) could be yielded, which could promote oxidative stress(Fu et al., 2018). We have add it in discuss. (line379-391)

  1. The results require a more thorough discussion, substantiated by relevant published materials.

Reply: We rewrited the discussion and added the relavant references.

Reviewer 3 Report

Shen and colleagues used Tibetan minipigs to study the serum metabolites during the occurrence and development of AS. It is a very interesting study, congratulations! However, the manuscript is defective in a few points which should be modified:

1- Some periods should be removed from the results and included in the discussion (i.e. lines 161-163, 180-186, 188-191, 210-213). The results section should illustrate the data obtained in this study.

2- The discussion is very succinct, and partially takes into account the data generated with the animal model. You need to expand the discussion by including more references to the clinical field if you want to suggest possible biomarkers of AS.

The quality of the English is good, minor corrections needed.

Author Response

Dear Editor and reviewers,

Thank you for your letter dated August 28. We were pleased to know that our work was rated as potentially acceptable for publication in Journal, subject to adequate revision. We thank the reviewers for the time and effort that they have put into reviewing the previous version of the manuscript. Their suggestions have enabled us to improve our work. Based on the instructions provided in your letter, we uploaded the file of the revised manuscript. Accordingly, we have uploaded a copy of the original manuscript with all the changes highlighted by using the track changes mode in Word.

Appended to this letter is our point-by-point response to the comments raised by the reviewers. The comments are reproduced and our responses are given directly afterward.

We would like also to thank you for allowing us to resubmit a revised copy of the manuscript.

We hope that the revised manuscript is accepted for publication in the Nutrients.

Reviewer 3

1. Some periods should be removed from the results and included in the discussion (i.e. lines 161-163, 180-186, 188-191, 210-213). The results section should illustrate the data obtained in this study.

Reply: We've moved into position. (line337-342, 369-378)

2. The discussion is very succinct, and partially takes into account the data generated with the animal model. You need to expand the discussion by including more references to the clinical field if you want to suggest possible biomarkers of AS.

Reply: We had modified the discussion and we discussed the possible biomarkers of AS in the clinical field.(line347-354)

Round 2

Reviewer 1 Report

The manuscript has been improved and all points raised were addressed.

One minor further comment:

The authors should only further tune down their tone with regards to translational aspects. The study was conducted on pigs and they were selected and not derived from the same litter. So, the discussion and conclusion should be not speculative but focussed on the results yielded.

Author Response

Dear Editor and reviewers,

Thank you for your letter dated September 15 . We were pleased to know that our work was rated as potentially acceptable for publication in Journal, subject to adequate revision. We thank the reviewers for the time and effort that they have put into reviewing the previous version of the manuscript. Their suggestions have enabled us to improve our work. Based on the instructions provided in your letter, we uploaded the file of the revised manuscript. Accordingly, we have uploaded a copy of the original manuscript with all the changes highlighted by using the track changes mode in Word.

Appended to this letter is our point-by-point response to the comments raised by the reviewers. The comments are reproduced and our responses are given directly afterward.

We would like also to thank you for allowing us to resubmit a revised copy of the manuscript.

We hope that the revised manuscript is accepted for publication in the Nutrients.

Reviewer 1

Comments to the Author

Here are the general comments from the reviewer.

The manuscript has been improved and all points raised were addressed.

Reply: Thank you for your comments and affirmations on this manuscript. Your comments make this manuscript more complete.

One minor further comment:

The authors should only further tune down their tone with regards to translational aspects. The study was conducted on pigs and they were selected and not derived from the same litter. So, the discussion and conclusion should be not speculative but focussed on the results yielded.

Reply: We are aware of this problem and we have made several changes to the discussion section to correct this problem. (line317, 327-329, 345-357, 373-374)

Reviewer 2 Report

The authors have appropriately addressed my concerns and revised the manuscript accordingly. The quality of the manuscript increased compared to its previous version. 

Author Response

Dear Editor and reviewers,

Thank you for your letter dated September 15 . We were pleased to know that our work was rated as potentially acceptable for publication in Journal, subject to adequate revision. We thank the reviewers for the time and effort that they have put into reviewing the previous version of the manuscript. Their suggestions have enabled us to improve our work. Based on the instructions provided in your letter, we uploaded the file of the revised manuscript. Accordingly, we have uploaded a copy of the original manuscript with all the changes highlighted by using the track changes mode in Word.

Appended to this letter is our point-by-point response to the comments raised by the reviewers. The comments are reproduced and our responses are given directly afterward.

We would like also to thank you for allowing us to resubmit a revised copy of the manuscript.

We hope that the revised manuscript is accepted for publication in the Nutrients.

Reviewer 2

Comments to the Author

Here are the general comments from the reviewer.

The authors have appropriately addressed my concerns and revised the manuscript accordingly. The quality of the manuscript increased compared to its previous version.

Reply: Thank you for your comments and affirmations on this manuscript. Your comments make this manuscript more complete.